# Time-frequency analysis reveals an association between the specific nuclear magnetic resonance (NMR) signal properties of serum samples and arteriosclerotic lesion progression in a diabetes mouse model

Kanako Yui[1], Yoshimasa Kanawaku[2]*, Akio Morita[3], Keiko Hirakawa[4], Fanlai Cui[2]

1 Division of Neurosurgery, Graduate School of Medicine, Nippon Medical School, Bunkyo-ku, Tokyo, Japan,
2 Department of Legal Medicine, Graduate School of Medicine, Nippon Medical School, Inzai, Chiba, Japan,
3 Geriatric Healthcare Center, Department of Neurosurgery, Teraoka Memorial Hospital, Fukuyama, Hiroshima, Japan, 4 Research Laboratory of Magnetic Resonance, Collaborative Research Center, Nippon Medical School, Bunkyo-ku, Tokyo, Japan

* ykanawaku@nms.ac.jp

## Abstract

Diabetes causes arteriosclerosis, primarily due to persistent hyperglycemia, subsequently leading to various cardiovascular events. No method has been established for directly detecting and evaluating arteriosclerotic lesions from blood samples of diabetic patients, as the mechanism of arteriosclerotic lesion formation, which involves complex molecular biological processes, has not been elucidated. "NMR modal analysis" is a technology that enables visualization of specific nuclear magnetic resonance (NMR) signal properties of blood samples. We hypothesized that this technique could be used to identify changes in blood status associated with the progression of arteriosclerotic lesions in the context of diabetes. The study aimed to assess the possibility of early detection and evaluation of arteriosclerotic lesions by NMR modal analysis of serum samples from diabetes model mice. Diabetes model mice (BKS.Cg db/db) were bred in a clean room and fed a normal diet. Blood samples were collected and centrifuged. Carotid arteries were collected for histological examination by hematoxylin and eosin staining on weeks 10, 14, 18, 22, and 26. The serum was separated and subjected to NMR modal analysis and biochemical examination. Mice typically show hyperglycemia at an early stage (8 weeks old), and pathological findings of a previous study showed that more than half of mice had atheromatous plaques at 18 weeks old, and severe arteriosclerotic lesions were observed in almost all mice after 22 weeks. Partial least squares regression analysis was performed, which showed that the mice were clearly classified into two groups with positive and negative score values within 18 weeks of age. The findings of this study revealed that NMR modal properties of serum are associated with arteriosclerotic lesions. Thus, it may be worth exploring the possibility that the risk of cardiovascular events in diabetic patients could be assessed using serum samples.

**Data Availability Statement:** All relevant data are within the paper and its Supporting information files.

**Funding:** The author(s) received no specific funding for this work.

**Competing interests:** The authors have declared that no competing interests exist.

# Introduction

## Background

Diabetes can cause arteriosclerosis, primarily due to persistent hyperglycemia, and arteriosclerosis is associated with subsequent development of various cardiovascular diseases [1, 2]. The detection of arteriosclerotic lesions at an early stage, before progression to a pathological condition with a high risk of cardiovascular events, is important in diabetic patient management. To diagnose a disease or pathological condition, it is important to determine the specific properties of the disease or pathological condition from clinical specimens such as serum. In a clinical setting, other tests (e.g., imaging [3–5] or functional tests [6, 7]) are needed in addition to blood tests to detect arteriosclerotic lesions. However, in Japan, there are no clear criteria for the screening of arteriosclerotic lesions, even though comprehensive risk management is recommended in clinical practice guidelines [8, 9].

There are many reports [10–14] on the changes in specific components of blood associated with arteriosclerosis and cardiovascular events in patients with diabetes mellitus. The ability to evaluate arteriosclerotic lesions using blood alone could make the management of arteriosclerosis more practical in clinical settings. Unfortunately, no tests have been established that can directly detect and evaluate arteriosclerotic lesions solely from blood samples of patients with diabetes mellitus, as the detailed mechanism of arteriosclerotic lesion formation, which involves complex molecular biological processes, has not been elucidated [15–17].

Serum contains many hydrogen atoms that make up water and other molecules. The interactions within and between molecules caused by hydrogen atoms affect the structure and function of molecules contained in serum. Hirakawa et al. developed an "NMR modal analysis" method [18] that enables visualization of the specific nuclear magnetic resonance (NMR) signal properties of a biological sample by measuring hydrogen atoms and analyzing them using the concepts and methods of modal analysis in vibration engineering. Data indicate that clinical specimens such as serum analyzed using this method exhibit different "NMR modes" depending on the presence of a disease or other pathological condition. As an example, there are published studies in which NMR modal analysis of serum from pancreatic cancer patients yielded results of diagnostic importance [19]. Therefore, we considered the possibility of applying our method not only to pancreatic cancer, but also to other diseases that indirectly or directly alter the physical properties of serum, specifically, the mobility of all hydrogen nuclei contained in serum. In the present study, we undertook an examination of the feasibility of NMR modal analysis for the evaluation and management of arteriosclerotic lesions associated with diabetes mellitus.

As patients with diabetes mellitus are multifactorial in terms of lifestyle, environment, and heredity [20, 21], it is difficult to group patients for studies of the specific NMR signal properties of arteriosclerotic lesions. For that reason, in the present study we utilized an animal model of diabetes with clearly identified risk factors in order to verify whether NMR modal analysis would be useful for the evaluation of arteriosclerotic lesions in humans.

## Objective

We conducted an arteriosclerotic lesion model experiment. The animal group was constructed using a diabetes mouse model of hyperglycemia, a risk factor for diabetes mellitus, from an early age. This study aimed to explore the possibility of early detection and evaluation of arteriosclerotic lesions from serum samples by NMR modal analysis using a diabetes mouse model with rapid onset and progression of arteriosclerotic lesions.

## Method

All mouse model protocols were in accordance with National Institute of Health guidelines. This study was approved by the Institutional Animal Care and Use Committee of our institution (permission number: 2020–094) and carried out according to the Nippon Medical School Animal Experimentation Regulations.

### Breeding of experimental animals

Diabetic model mice (BKS.Cg db/db) and control mice (Jcl:ICR) were purchased from Nippon Crea Co., Ltd., Tokyo, Japan, and kept in a clean room in the Laboratory Animal Care Room of Nippon Medical School. The diet consisted of general solid food (MF from Oriental Yeast Co., Ltd., Tokyo, Japan), and the animals were allowed to eat and drink freely.

### Understanding of blood glucose dynamics

Glucose tolerance testing and blood glucose monitoring were performed on a set of 16 mice (4 mice each of the Jcl:ICR, C57BL/6JJcl, BKS.Cg db/db, and BKS.Cg -/db strains) during rearing to determine their blood glucose dynamics. Glucose tolerance tests were conducted periodically from the age of 8 weeks. After fasting for 12 h from the previous day, 2 g/kg of 50% glucose was administered, and blood glucose was measured at 0 min, 30 min, 1 h, and 2 h after administration using a commercially available rapid kit for humans (Safe-ACCU, Sinocare, No.265 Guyuan Road Hi-tech Zone, Changsha, Hunan, China). The kit was unable to take measurements when blood glucose exceeded 600 mg/dL. At 8 weeks of age, the fasting blood glucose exceeded 200 mg/dL, which made measurement with the rapid kit difficult when glucose was loaded. Blood glucose was also measured sporadically, and the animals were confirmed to be hyperglycemic at each week of life. No clear sex differences in blood glucose measurements were observed at any week or in any lineage.

### Weeks of sampling and sample size

Based on the hemodynamic parameters described in the previous section, we estimated 10–26 weeks to be the age at which hyperglycemia was maintained, making that an appropriate time to evaluate changes in serum data and pathological findings. Samples were taken every 4 weeks for 26 weeks, starting at 10 weeks of age for BKS.Cg db/db mice, apart from the aforementioned set of mice.

We initially chose a small sample size for the first analysis of mouse serum using NMR modal analysis. Specifically, two males and two females were selected for each 1-week-old group of BKS.Cg db/db mice. However, we later increased the sample size for 1-week-old mice with rapidly progressing lesions to better identify atherosclerotic lesions, which was the objective of our study. The total number of mice sampled was 28 BKS.Cg db/db (4 mice each at 10, 14, 22, and 26 weeks of age and the remaining 12 mice at 18 weeks of age) and 7 Jcl:ICR mice.

### Sample collection and tests performed

The animals were euthanized by opening the chest and drawing cardiac blood (approximately 1 mL) after maintenance anesthesia with isoflurane to obtain adequate analgesia.

Collected blood was centrifuged, and serum was aliquoted for the biochemistry testing and NMR analysis. Samples for NMR measurement were stored at −80˚C until analysis [22]. Samples for blood biochemical tests were selected from one male and one female in each group of mice. In addition to blood glucose, total cholesterol (T-Cho) and triglyceride (TG) levels were

measured as serum components related to arteriosclerotic lesions. Measurements were out-sourced to Oriental Yeast Co. (Shiga, Japan).

## Pathological testing

After blood collection, carotid arteries were collected for histological examination by hematoxylin and eosin (H&E) staining. Carotid specimens for pathological examination could not be obtained from specimen no. 22–2 (22 weeks old) due to difficulty in identifying the carotid artery. After collection, the specimens were immediately fixed in formalin solution. Processing after embedding and H&E staining were outsourced to Genostaff Co. (Tokyo, Japan).

## Free induction decay data collection for NMR measurement

Each serum sample (100 μL) was mixed with 500 μL of deuterium oxide ($^2H_2O$) (ISOTEC, Sigma-Aldrich, St. Louis, MO, USA) and pipetted into a 5-mm (outer diameter) NMR tube (Wilmad-LabGlass, Vineland, NJ, USA) for NMR analysis [23]. Solution-state NMR analyses were performed at a proton resonance frequency of 400 MHz (9.4 Tesla) using an ECZ™ NMR spectrometer (JEOL Ltd., Tokyo, Japan) interfaced with a probe (digital auto-tunable type [NM-03812RO5S]) and equipped with Delta™ NMR processing and control software, version 5.3.2 (JEOL Ltd.). The field was locked to the $^2H$ resonance of the $^2H_2O$ solvent. One-dimensional proton NMR signals were automatically acquired at a probe temperature of 30˚C using the program supplied by JEOL that supported the macro function in Delta™. Free induction decay (FID) data were acquired using a single pulse with a 2.0-s relaxation delay between repeated pulse sequences. The strong signal arising from free water was suppressed using DANTE presaturation. Other conditions were as follows: radio-frequency pulse width, 2.93 μs; acquisition time, 1.636 s; repetition time, 3.636 s; spectral width, 10,016 Hz; number of data points, 16,384; and number of steady-state transients, 400. The FID data were saved in JEOL Delta format (JDF).

## NMR data analysis

For time-frequency analysis, which analyzed how the frequency content of FIDs changed over time, short-time Fourier transform (STFT) was used. The STFT was calculated using a self-made application program ("STFT tool") developed for MATLAB™ (The MathWorks, Inc., Natick, MA, USA), a programming and numeric computing platform.

The FID data for each serum sample were imported into the STFT tool, and STFT processing and spectrogram calculation were performed. The spectrogram data were output as images (TIFF format) and ASCII data. The STFT and spectrogram calculations were performed using a previously reported algorithm [24].

The STFT matrix was given by X (f) = [X$_1$ (f) X$_2$ (f) X$_3$ (f) X$_k$ (f)], such that the $m$th element of the matrix was:

$$X_m(f) = \sum_{n=\infty}^{\infty} x(n)g(n - mR)e^{-j2\pi fn}$$

In this formula, $g$ (n) represents the window function of length $M$, X$_m$ (f) represents the DFT of windowed data centered about time $mR$, and $R$ represents Hop size (the difference between the window length $M$ and the overlap length $L$) between successive DFTs.

The number of columns in the STFT matrix was given by:

$$k = \frac{N_x - L}{M - L}\rfloor$$

In this formula, $N_x$ represents the length of the original signal $x$ (n), and the $\lfloor \rfloor$ symbols denote the floor function. The number of rows in the matrix equaled NDFT, the number of DFT points, for centered and two-sided transforms and $\lfloor NDFT/2 \rfloor + 1$ for one-sided transforms.

When a nonzero overlap length $L$ was specified, it was represented by $R = M-L$. $R$ varied from 0 to $M - 1$. $f$ represents the index spanning and varies from 0 to (FFT block size $- 1$). The parameters used in this study were as follows: $g$ hanning, $L$ 64, $R$ 32, FFT block size 1024 (= 1/$n$), and $M$ 256. A spectrogram was generated as a visual representation of the time-frequency analysis. The data for the STFT-based spectrogram, $P_{SP}$, were calculated by the square of $X(f)$. Each spectrogram datum in the two-dimensional matrix was read from left to right and row by row, populated in a single row. In this study, each spectrogram datum was reshaped from a 256×1024 two-dimensional matrix into a 1×262,144 single row. After combining the total rows into a separate matrix, (number of samples)×262,144, the resulting dataset was used for multivariate data analysis.

## Multivariate analysis

Similar to the study of the diagnosis of pancreatic cancer from serum by applying NMR modal methods [19], pattern recognition [25] of image data was applied to clarify groups from [1]H NMR-FID spectrograms of mouse serum in this study, and principal component analysis (PCA) and partial least squares discriminant analysis (PLS-DA) were performed. To perform PCA and PLS-DA, all individual spectrogram data were imported into the Unscrambler X software (version 11; Camo Software AS, Oslo, Norway). After importation, the data were pre-processed by Standard Normal Variates transformation (Standard Normalization).

PCA is a method of dimensionality reduction of data using only explanatory variables and can be used for exploratory analyses such as identifying outliers and potential subgroups [26]. In this study, the PCA method was used to examine the relationship between the principal component score of each individual and the age in weeks and pathological findings [27]. Based on the results of the PCA, we regrouped the spectrogram data according to age in weeks and the progression of atherosclerotic lesions. We then analyzed the spectrogram data using PLS-DA [28–30] to visualize the presence or absence of differences between the regrouped mice and the time-frequency regions of the NMR signals associated with the differences.

The NMR signal regions associated with the differences between groups were considered the key variables, and to visualize these regions, the correlation loading values for each factor were plotted on the spectrogram. This was referred to as a "correlation loading plot".

The score plot was displayed using the Unscrambler drawing function, and the correlation loading plot was displayed using a program written for MATLAB. Variables with an absolute value $\geq 0.4$ as the loading value were considered relevant.

A graphical flow chart from the animal experiments to the multivariate analysis performed in this study is shown as supporting information (S1 and S2 Figs).

## Results

### Pathological results

Carotid artery pathology was observed, and Table 1 lists the lesions observed at each week for each mouse of the Jcl:ICR and BKS.Cg db/db strains, except for mouse ID.B22-2, from which

**Table 1. Summary of Jcl:ICR and BKS.Cg db/db carotid artery lesion evaluation.**

|  | Week | Sex | Subject ID | No lesion | Diffuse intimal thickening | Atheroma | Severity |
|---|---|---|---|---|---|---|---|
| Jcl:ICR | 18 | male (n = 3) | J18-1,2,3 | 0 | 3 | 0 | Moderate |
|  |  | female (n = 4) | J18-4,5,6,7 | 0 | 4 | 0 | Moderate |
| BKS | 10 | male (n = 2) | B10-1 | 0 | 1 | 0 | Moderate |
|  |  |  | B10-2 | 1 | 0 | 0 | Mild |
|  |  | female (n = 2) | B10-3, 4 | 0 | 2 | 0 | Moderate |
|  | 14 | male (n = 2) | B14-1,2 | 0 | 2 | 0 | Moderate |
|  |  | female (n = 2) | B14-3,4 | 0 | 2 | 0 | Moderate |
|  | 18 | male (n = 6) | B18-3 | 0 | 1 | 0 | Moderate |
|  |  |  | B18-1,2,4,5,6 | 0 | 5 | 5 | Severe |
|  |  | female (n = 6) | B18-8,9,10,11 | 0 | 4 | 0 | Moderate |
|  |  |  | B18-7,12 | 0 | 2 | 2 | Severe |
|  | 22 | male (n = 2) | B22-1 | 0 | 1 | 1 | Severe |
|  |  |  | B22-2 | none | none | none |  |
|  |  | female (n = 2) | B22-3 | 0 | 1 | 0 | Moderate |
|  |  |  | B22-4 | 0 | 1 | 1 | Severe |
|  | 26 | male (n = 2) | B26-1,2 | 0 | 2 | 2 | Severe |
|  |  | female (n = 2) | B26-3,4 | 0 | 2 | 2 | Severe |

carotid artery samples could not be obtained at necropsy. For Jcl:ICR, observations were only made at 18 weeks of age, and although all mice exhibited mild diffuse intimal thickening, no atheromatous plaques were observed. In the BKS.Cg db/db mice, diffuse intimal thickening, an indicator of moderate atherosclerotic lesions, was observed in 3 of 4 mice at 10 weeks of age and in all mice after 14 weeks of age. No atherosclerotic plaques were observed in any of the mice at 10 or 14 weeks of age. At 18 weeks of age, plaques were present in 5 of 6 males and 2 of 6 females. At 22 and 26 weeks, the presence of atherosclerotic plaques was confirmed in all observed mice.

The presence or absence of 3 items of carotid artery pathological findings in the two mouse strains, Jcl:ICR and BKS.Cg db/db, was tabulated for each week of age. The degree of progression of atherosclerotic lesions was judged as moderate for diffuse intimal thickening, severely advanced for atheromatous plaques, and mild for none of the above.

In the Jcl:ICR sample, although diffuse intimal thickening was indicated to some extent, none of the samples exhibited marked thickening, and none had atheromatous plaques.

In BKS.Cg db/db samples, mice already exhibited intimal thickening at 10 weeks of age, and many more mice developed atheromaatous plaques after 18 weeks of age. In BKS.Cg db/db, individuals with atheromatous plaques appeared after 18 weeks of age. The proportion increased stepwise as age increased. In mouse B22-2 only, the carotid artery could not be identified and collected for histological examination.

Representative H&E staining of the carotid arteries of BKS.Cg db/db mice is shown in Fig 1 (A) B10-1, Fig 1(B) B14-4, and Fig 1(C) B18-5. Fig 1D shows a photograph of H&E staining of J18-1. BKS.Cg db/db mice exhibited clear exacerbation of the degree of intimal thickening at the ages of 10, 14, and 18 weeks, as atherosclerosis progressed over a short period. At 18 weeks of age, BKS.Cg db/db mice exhibited a greater degree of intimal thickening than Jcl:ICR mice at 18 weeks of age, and the presence of atheromatous plaques was evident in more than half of the animals, indicating that atherosclerosis progressed rapidly at this age. At 18 weeks of age,

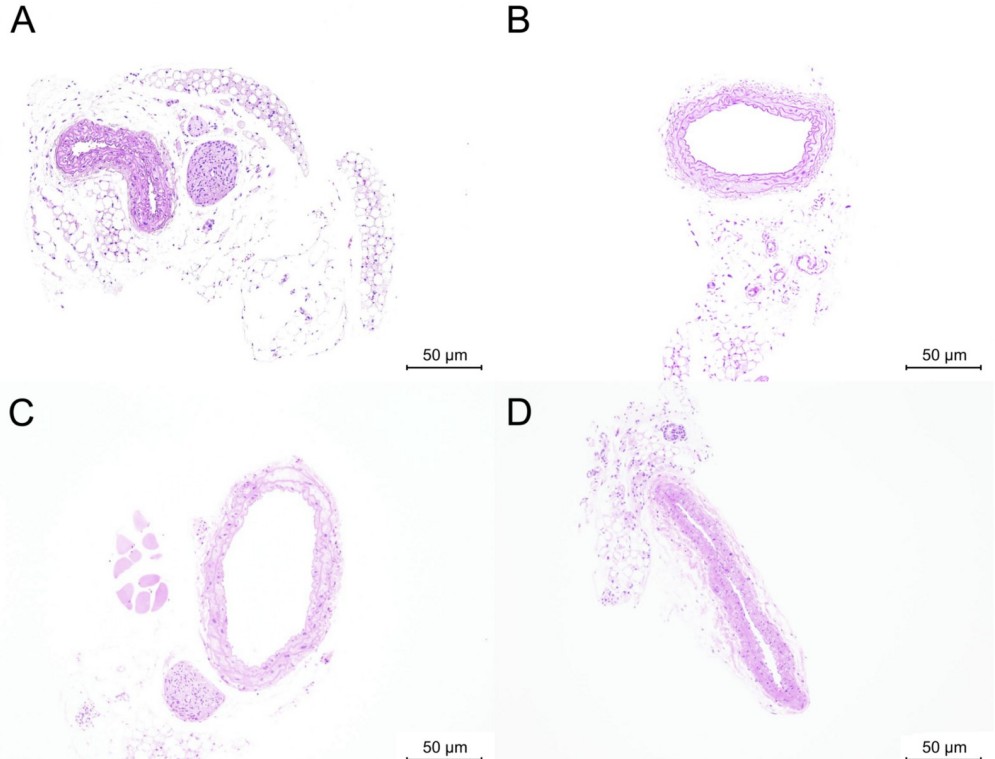

**Fig 1. H&E staining of carotid artery samples.** (A) Example of H&E staining of a carotid artery specimen from a 10-week-old BKS.Cg db/db mouse (B10-1), showing normal histology, with the tunica media composed of numerous smooth muscle cells. (B) Example of H&E staining of a carotid artery specimen from a 14-week-old BKS.Cg db/db mouse (B14-4), showing progressive intimal thickening compared with (A). (C) H&E staining of a carotid artery specimen from an 18-week-old BKS.Cg db/db mouse (B18-5), showing a more advanced disease stage and atheromatous plaques. (D) H&E staining of a carotid artery specimen from an 18-week-old Jcl:ICR mouse (J18-1) showing intimal thickening.

there was a difference between males and females in terms of the presence or absence of atheromatous plaques.

## Blood biochemistry tests

Table 2 shows the results of biochemical tests performed on two mice during each week. Blood glucose levels exceeded 600 mg/dL in all BKS.Cg db/db mice. Conversely, T-Cho values were generally slightly higher in BKS.Cg db/db mice than in Jcl:ICR mice, but they did not differ by week. Neutral fat values varied depending on the previous feeding condition, and under free-feeding conditions, there was a large variation among individuals, and no strain or week-of-age–related characteristics were confirmed.

## NMR modal analysis

Fig 2 shows a representative spectrogram of specimen B18-1. Although all spectrogram images were aligned and visually observed and compared with respect to strain, age, and pathological findings, identifying individual characteristics was difficult.

The PCA on spectrogram data of all mouse serum samples showed that the BKS and control groups clustered in the PC1/PC2 score plots (data not shown).

**Table 2. Results of biochemical tests performed on two BKS.Cg db/db mice at each week of age.**

|  | No. | BS (mg/dL) | T-Cho (mg/dL) | TG (mg/dL) |
|---|---|---|---|---|
| BKS.Cg db/db 10W | B10-1 | 846 | 146 | 328 |
|  | B10-3 | 818 | 118 | 146 |
| BKS.Cg db/db 14W | B14-1 | 681 | 131 | 196 |
|  | B14-3 | 796 | 155 | 217 |
| BKS.Cg db/db 18W | B18-2 | 647 | 132 | 86 |
|  | B18-9 | 766 | 134 | 210 |
| BKS.Cg db/db 22W | B22-2 | 838 | 154 | 182 |
|  | B22-3 | 1004 | 92 | 90 |
| BKS.Cg db/db 26W | B26-1 | 832 | 141 | 122 |
|  | B26-4 | 867 | 107 | 94 |
| Jcl:ICR 18W | J18-2 | 127 | 130 | 207 |
|  | J18-6 | 84 | 86 | 56 |

An elevated blood glucose level exceeding 600 mg/dL was observed in both weeks, whereas total cholesterol was only slightly higher than in Jcl:ICR mice.

Triglyceride levels varied depending on the previous feeding situation in both the Jcl:ICR and BKS.Cg db/db groups.

The results of PCA on the spectrogram data for serum samples from all BKS.Cg db/db mice are shown in Fig 3. In this figure, the PC1/PC5 score plot (Fig 3A) showed that groups B10 and B14 formed one loose cluster with similar PC1 scores. PC1 scores were organized as follows: B26 group > B22 group > B10 and B14 groups. In contrast, the PC1 scores of the B18 group varied widely from individual to individual, making it difficult to characterize the B18 group as a single population. On the score plot, we observed two subgroups: six mice with negative PC1 scores, similar to the B10 and B14 groups (B18-2, B18-3, B18-4, B18-6, B18-9, and B18-11), and six mice with positive PC1 scores, similar to the B22 and B26 groups (B18-1, B18-5, B18-7, B18-8, B18-10, and B18-12). Fig 3B shows a correlation loading plot of PC1. The regions of different positive and negative loading values are clearly displayed in the NMR signal-derived regions observed in the spectrogram of Fig 2, indicating that the PC1 scores reflect differences in the time-frequency characteristics of the NMR signals of the samples.

The results of PLS-DA in groups B10 and B14 and groups B22 and B26, which had a tendency to form clusters, are shown in Fig 4. In the Factor-1/Factor-2 score plot (Fig 4A), groups B10 and B14 and groups B22 and B26 were roughly divided by Factor-1 into clear clusters, with the B10 and B14 groups forming a cluster with similar scores and the B22 and B26 groups, except for B26-1 and B26-2, also forming a loose cluster. Fig 4B displays a correlation loading plot of Factor-1, showing the presence of variables important for the separation of the two groups on the time-frequency domain of the NMR signal. Because PLS-DA was used as a 2-class classifier in the present study, Factor-1 in Fig 4 is interpreted as an axis that reflects group information; axes from Factor-2 onward are interpreted as axes that maximize the separation on Factor-1 and make it visible on the plot. This is also the case for Figs 5 through 7. This interpretation applies to Figs 5–7 as well.

Fig 5 shows the results of the PLS-DA performed on the aforementioned PCA, distinguishing six mice from group B18 with negative PC1 scores, similar to groups B10 and B14, and mice from group B18 with positive PC1 scores, similar to groups B22 and B26. Negative value samples are denoted as 18W A, and positive value samples are denoted as 18W B. In the Factor-1/Factor-2 score plot (Fig 5A), the results are dichotomized by Factor-1. Fig 5B is a

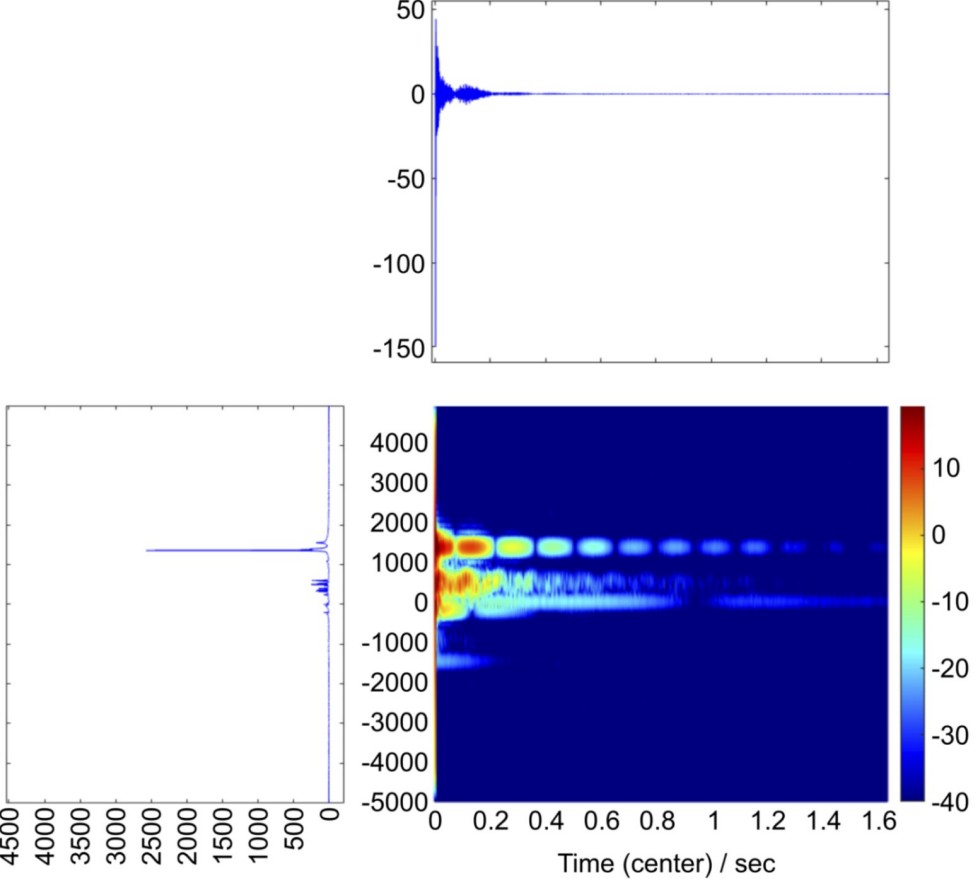

**Fig 2. Representative spectrogram of an 18-week-old BKS.Cg db/db mouse (B18-1).** The x-axis indicates time (s), and the y-axis indicates frequency (Hz). The conditions for STFT were as follows: FFT points, 1024; separate points, 512; shift length, 32; window length, 64.

correlation loading plot for Factor-1, showing the presence of variables important for the separation of the two groups on the time-frequency domain of the NMR signal.

PLS-DA was performed by combining groups B10, B14, and B18 A and groups B22, B26, and B18 B, respectively, under the assumption that the total sample was dichotomized within the 18-week-old group. The results are shown in Fig 6, in which groups B10, B14, and B18 A are labeled as precursor groups and groups B22, B26, and B18 B as progression groups. In the Factor-1/Factor-2 score plot (Fig 6A), the results were dichotomized by Factor-1. Fig 6B is a correlation loading plot of Factor-1, showing the presence of variables important for the separation of the two groups on the time-frequency domain of the NMR signal.

As shown in Fig 6, although six mice from groups B10, B14, and B18 A grouped together as a precursor group, as in the first PCA method, the score values of these groups differed slightly. Fig 7 displays the results of PLS-DA with respect to these data. In the Factor-1/Factor-3 score plot (Fig 7A), Factor-1 dichotomizes the two groups into B10, B14, and B18 A groups. Fig 7B is a correlation loading plot of Factor-1, showing the presence of variables important for the separation of the two groups on the time-frequency domain of the NMR signal.

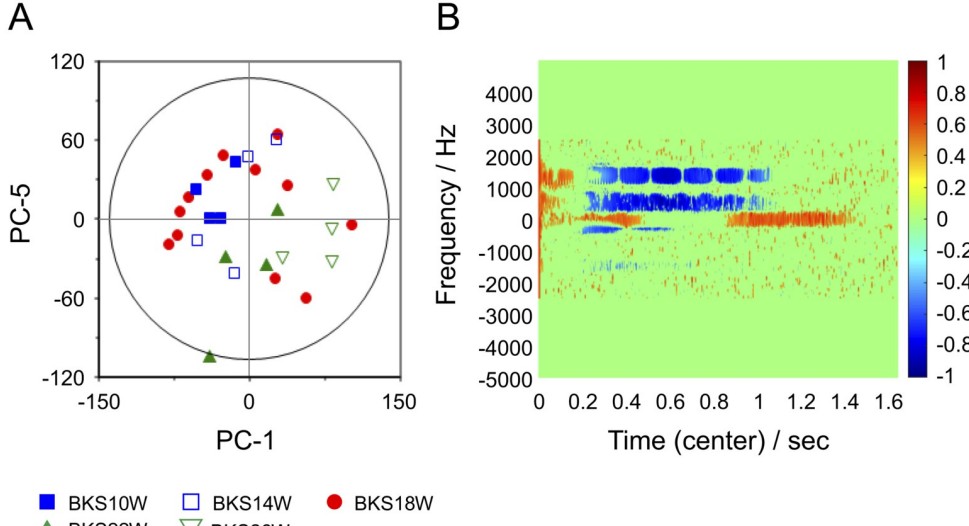

**Fig 3. Results of principal component analysis using spectrogram data from all BKS.Cg db/db serum samples.** (A) PCA PC-1/PC-5 score plot diagram. Principal component 1 score was interpreted to be positively associated with week-old progression, whereas principal component 5 score was interpreted to be negatively and moderately associated with week-old progression. Furthermore, on closer examination of principal component 1, it shows a tendency to group with different score values for 10- and 14-week-old and 22- and 26-week-old mice. The 18-week-old mice are not uniform in score values but dispersed, with scores approximating the two groups. Furthermore, although the blood biochemistry test results shown in Table 2 varied from individual to individual, there is no clear correlation between this variation and the variation in the PCA score plots in Fig 3. (B) Principal component 1 loading diagram. The relative loading data are mapped onto the spectrogram area. The darker red and blue areas indicate the frequency components that are significantly related to the positive and negative contribution of each principal component. The threshold values, which were set in an exploratory manner based on the characteristics of the analyzed data to increase discriminability, were set above 0.4 and below −0.4. Values closer to 1 are shown in darker red, and values closer to −1 are shown in darker blue.

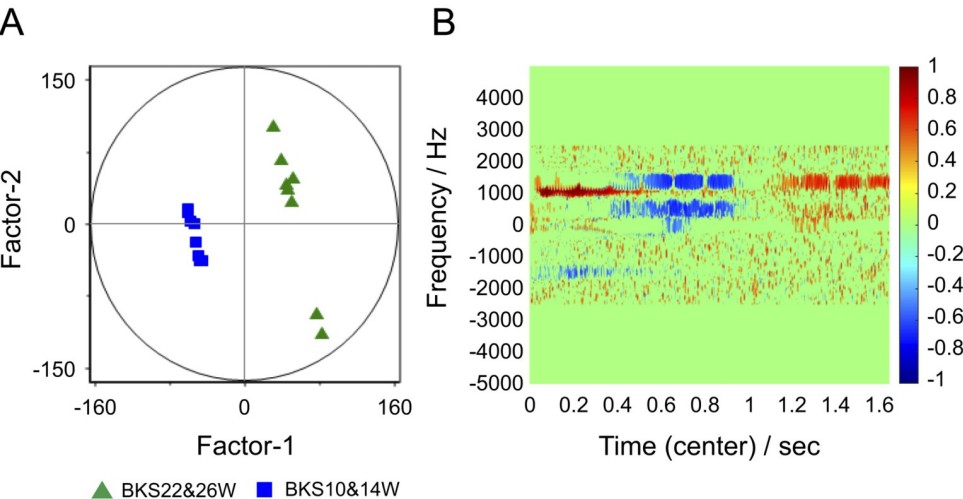

**Fig 4. Results of partial least squares discriminant analysis performed on spectrogram data for serum samples from 10- and 14-week-old and 22- and 26-week-old mice that grouped as shown in Fig 3A.** (A) Score plot of Factor-1/Factor-2 of PLS-DA. Factor-1 is clearly divided into two groups at 10 and 14 weeks of age and at 22 and 26 weeks of age. (B) Factor-1 loading diagram.

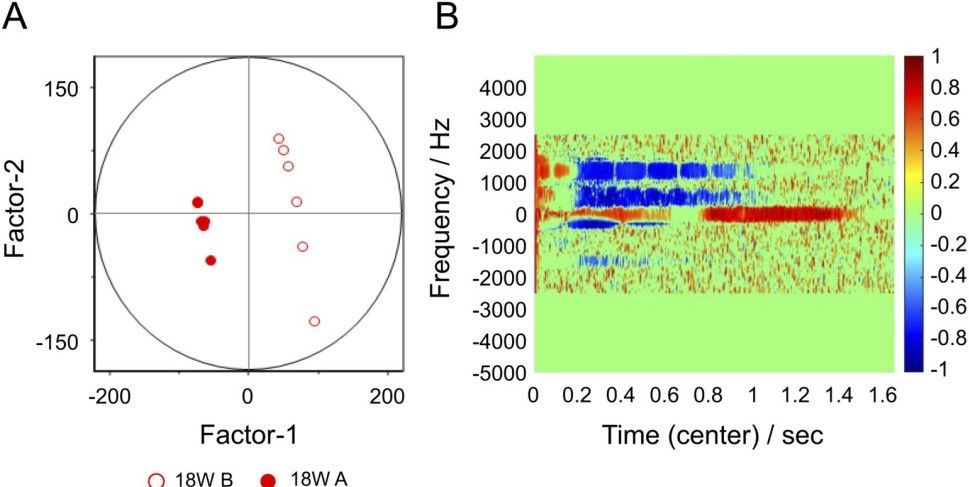

**Fig 5. Results of partial least squares discriminant analysis performed on spectrogram data for serum samples from mice at 18 weeks of age only.** (A) Factor-1/Factor-2 score plot. Samples bisected by PCA in Fig 5 and those on the 10- and 14-week side are labeled the 18W A group and those on the 22- or 26-week side are labeled the 18W B group. Samples are clearly dichotomized by Factor-1 at 18W A and 18W B. (B) Factor-1 loading diagram.

## Discussion

The results of this study showed that NMR modal properties of serum samples resulting from analysis of NMR-FID signals were associated with arteriosclerotic lesions. In addition, this study successfully showed that NMR modal properties can vary as arteriosclerotic lesions progress. It can be pointed out that one of the factors contributing to this success was the successful completion of animal experiments conducted under well-controlled conditions, and with the exception of one individual, pathology experiments on mouse carotid arteries were

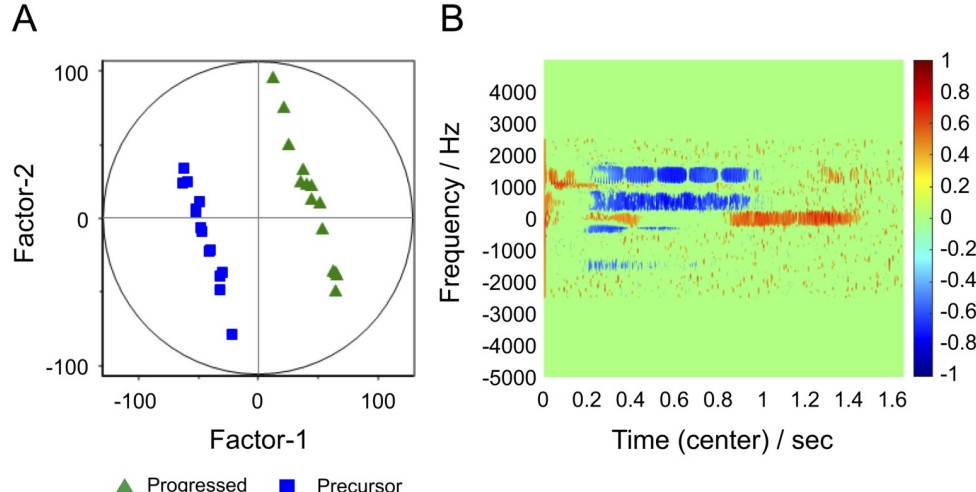

**Fig 6. Results of partial least squares discriminant analysis performed on spectrogram data for all serum samples of BKS.Cg db/db mice.** (A) Factor-1/Factor-2 score plot. Samples that were negatively distributed after PC-1 in Fig 3 are merged into the precursor group, and samples that were positively distributed are merged into the progressed group. Samples are clearly dichotomized by Factor-1. (B) Factor-1 loading diagram. The distribution is similar to the sum of Figs 6B and 7B.

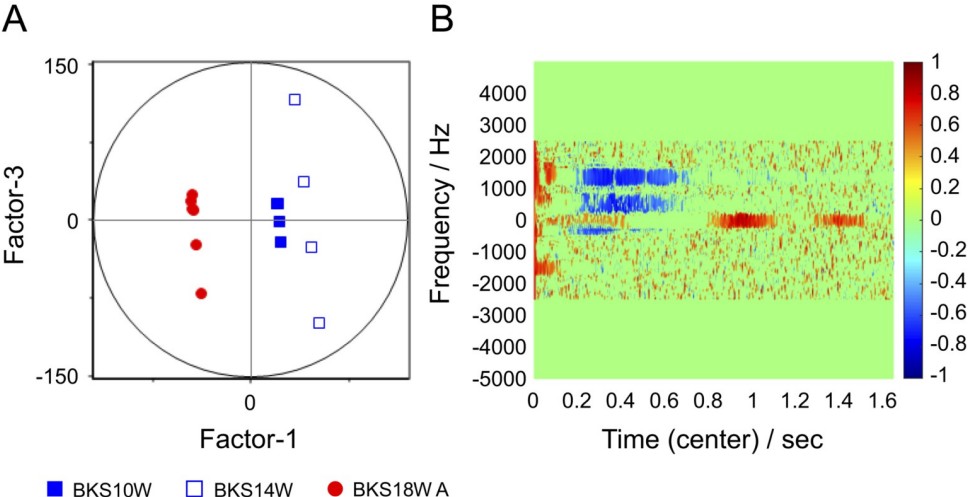

**Fig 7. Results of partial least squares discriminant analysis performed on spectrogram data of serum samples from 10-, 14-, and 18-week-old mice.** (A) Factor-1/Factor-3 score plot, with Factor-1 dichotomized between 18- and 10- and 14-week-old mice. (B) Factor-1 loading diagram. Some of the loading values seem to be similar to those in Fig 5B.

also successful. In fact, it has been pointed out that the first animal experiment is a very important element in the interpretation of the results of multivariate analysis in the step of chemometric studies [31].

Based on the score plots of all PCA data for BKS.Cg db/db mice, all groups (except B18) were divided into a precursor group and progression group by PC-1. In contrast, the data plots for group B18 were scattered and bifurcated into precursor and progression groups, as the arteriosclerotic lesions progressed particularly rapidly. If the scatter of the score plots in Fig 3 reflected differences in week age, a distribution by age group would be expected, but the actual results are contrary to this. Since PCA is an analysis that uses only explanatory variables (in this experiment, the serum spectrogram data) for dimensionality reduction [26], it was assumed that this result could be interpreted as a visualization of the change in serum status before and after 18 weeks of age.

Based on the results of PLS-DA (Fig 6A), the data plots for group B18 were ultimately divided into a precursor group and a progression group. Careful pathological examination showed that some samples from group B18 exhibited evidence of atheromatous plaques; part of these samples was classified into the precursor group. As shown in Fig 7A, the PLS-DA results for groups B10, B14, and B18 A showed that the samples were divided into these groups by Factor-1. The PLS-DA score plot for this precursor group shown in Fig 7A cannot be explained by age-related changes alone. In this study, time-trend changes in carotid artery pathology were observed, which, as noted above, showed rapid atherosclerotic lesion progression at 18 weeks. Considering the characteristics of the distribution of the B18 group shown on some score plots, we interpret this as suggesting that the clustering and separation of samples shown in the multivariate score plots in this study may be related to the progression of atherosclerotic lesions, though there is a certain degree of influence of differences in age.

In addition, the time-frequency domains of the variables valid for discrimination shown in the loading plots in Figs 3–7 corresponded roughly to the signal domains in the serum spectrograms in Fig 2. These results suggest that the progression of arteriosclerotic lesions was reflected in the serum NMR modal properties.

NMR, which enables analyses of the structure and energy state of matter at the atomic level, is an analytical method that examines interactions between nuclei and radiofrequency waves [32]. Medical applications of NMR include relaxometry and spectroscopy. Relaxometry measures and compares the relaxation time (time-series data analysis) of an FID signal, such as the basic technique of magnetic resonance imaging, which is widely used in clinical medicine. Spectroscopy is applied in the comprehensive chemical analysis of metabolites [33–37] by measuring and comparing spectra (frequency analysis). Spectroscopy is a basic research tool in the field of metabolomics, and several reports [38–42] have suggested its suitability for use in clinical diagnosis. The NMR modal analysis technique used in this study is based on the concept of "modal analysis" in vibration engineering [43, 44], which analyzes FID signal properties.

The present data suggest that this NMR technique is highly suitable for analyzing biomedical samples. Serum, which was targeted in this study, is a complex mixture containing considerable amounts of albumin in addition to numerous metabolites and small ionic molecules. Hydrogen atoms contained in water and other molecules in serum play significant roles in interactions within and between molecules. As such, spectrograms calculated from FID were prepared to visualize the properties of the NMR mode for an entire sample, which was determined by the interactions with protons within and between molecules. The correlation loading plots indicate which parts of the NMR modal properties are valid for discrimination. These plots can be interpreted as presenting a visualization of the parts of the overall properties of the serum sample that are relevant to the characteristics of the mouse model of atherosclerosis. Therefore, the results of this study suggest that specific NMR modal properties of serum are associated with the onset of arteriosclerosis due to hyperglycemia and that such properties can be identified using this approach. NMR metabolomics experiments focus on low molecular weight substances and obtain their spectra [45]. On the other hand, in NMR mode analysis, all molecular species including protons of up to high molecular weight substances such as serum proteins in addition to low molecular weight metabolites are measured by NMR in a non-selective and comprehensive manner to obtain spectrograms that reflect the state of all hydrogen atoms in the serum sample. This allows us to interpret that biochemical changes due to disease or pathological abnormalities may be reflected in not only static but also dynamic changes of protons in all components in serum (from low to high molecular weight substances and free water). In this study, it should be noted that, since atherosclerotic lesions are associated with certain biochemical changes in serum (inflammatory markers and hormones), the modal properties of this serum NMR should be considered to include inflammatory markers and hormonal changes associated with the progression of atherosclerotic lesions.

This study had some limitations. First, experimental animals were used in this study, so no human blood samples were analyzed. In addition, arteriosclerosis can be caused by multiple factors other than diabetes, such as hypertension and lipid abnormalities, but the mice used in this study had only persistent hyperglycemia. It is also known that the risk of cardiac and renal events increases as the number of risk factors increases and arteriosclerotic lesions progress [8–10, 16, 17, 46, 47]. Second, there was some discordance between the multivariate analysis results and pathological findings of BKS.Cg db/db mice based on differences in the biological reactions in individual mice. The biological reactions may partly reflect the serum of mice with atherosclerosis. On the other hand, in 18-week-old mice with rapidly progressive atherosclerosis, the reaction may be associated with pathological progression. Unfortunately, our exploratory studies are limited in their ability to elucidate the true nature of this biological response, and it will be the subject of future research. These limitations indicate that our results cannot be immediately extrapolated to humans.

Despite the above-mentioned limitations of this study, the arteriosclerosis model experiments using BKS.Cg db/db mice, which have persistent hyperglycemia, were successful. Continued hyperglycemia is diagnosed as diabetes in human clinical medicine, but there are no clear diagnostic criteria for diabetes in laboratory animals [48]. Thus, both glucose tolerance tests and occasional blood glucose measurements [49, 50] were performed in the present study, confirming that the mice had hyperglycemia before sexual maturity, which is considered indicative of diabetes in humans. Based on these findings, we consider that it may be worth exploring the possibility that arteriosclerosis, a risk factor for cardiovascular events in diabetic patients, can be assessed using NMR modal analysis of serum samples.

## Conclusions

The results of this study suggest that NMR modal analysis of serum samples obtained from mice with persistent hyperglycemia can be used to monitor the development of atherosclerosis. The mice used in the present study exhibited severe atherosclerotic lesions at 18 weeks of age, and serum samples from the borderline weeks showed differences in NMR modal data. We expect that further studies and appropriate modeling will enable clinical examination of the progression of diabetes-related atherosclerotic lesions using blood samples.

## Supporting information

**S1 Fig. Graphical flow chart from the animal experiments to the multivariate analysis.** In this figure, the results of NMR analysis of blood samples collected after animal experiments in groups A (n = 10) and B (n = 10) are shown. In total, 20 $^1$H-NMR FID signal data were acquired, which were short-time Fourier transformed [51], and 20 spectrograms were calculated. For multivariate analysis, a process was performed to obtain a data matrix from the 10 spectrograms. From this data matrix, an exploratory analysis was performed using principal component analysis [26] for the identification of outliers and potential subgroups of animals in groups A and B. From this figure, it was possible to identify subgroups that were not initially anticipated, so PLS-DA [52, 53] was performed only with data that corresponded to subgroups in the same data matrix to analyze whether subgroups could be identified. In both principal component analysis and PLS-DA, the variables that were important for the identification of groups in each analysis were plotted as corresponding score plots. This score plot has the same time and frequency axes as the spectrogram and thus can be compared with the spectrogram. The illustration of the mouse and the NMR equipment in this figure was obtained from Research Net, a website where research illustration material is available free of charge (https://www.wdb.com/kenq/illust $^©$ WDB Co., Ltd., Tokyo, Japan).
References (cited only in the Supporting Information)
51. Wacker M, Witte H. Time-frequency techniques in biomedical signal analysis. a tutorial review of similarities and differences. Methods Inf Med. 2013;52:279–296.
52. Barker M, Rayens W. Partial least squares for discrimination. J Chemo. 2003;17:166–173.
53. Sylvie Chevallier S, Bertrand D, Kohler A, Courcoux P. Application of PLS-DA in multivariate image analysis. J Chemo. 2006;20: 221–229.
(TIF)

**S2 Fig. Graphical flow chart of matrixing of spectrogram data.** The process surrounded by the green box in S1 Fig is described in detail. For multivariate analysis of spectrogram data, a $256 \times 1024$ spectrogram (a) with frequency on the horizontal axis and time on the vertical axis was divided into 1024 single rows (b). All rows were re-arranged into a single row, and $1 \times 262,144$ single rows were reshaped (c). Since a single row represents spectrogram data for

one individual (d), the single rows of spectrogram data for all individuals needed for multivariate analysis are combined to create a data matrix (e).
(TIF)

**S1 Data.**
(ZIP)

## Acknowledgments

We thank Haruo Hashimoto for guidance in the animal experiments, Kaori Okihara and Tomoko Konta for assistance with NMR measurements, and Hironobu Yoshimura, Naohito Ogiso, and Tomoki Nakao of JEOL, Ltd. (Tokyo, Japan) for providing technical assistance with the NMR data analysis.

## Author Contributions

**Conceptualization:** Kanako Yui, Yoshimasa Kanawaku, Akio Morita, Keiko Hirakawa, Fanlai Cui.

**Data curation:** Kanako Yui, Yoshimasa Kanawaku, Akio Morita, Keiko Hirakawa, Fanlai Cui.

**Formal analysis:** Kanako Yui, Yoshimasa Kanawaku, Keiko Hirakawa, Fanlai Cui.

**Investigation:** Kanako Yui.

**Writing – review & editing:** Kanako Yui, Yoshimasa Kanawaku.

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
