## [Decision Letter · Decision Letter 0]

12 Dec 2023

PONE-D-23-28528Time-frequency analysis reveals an association between the specific nuclear magnetic resonance (NMR) signal properties of serum samples and arteriosclerotic lesion progression in a diabetes mouse modelPLOS ONE

Dear Dr. Kanawaku,

Thank you for submitting your manuscript to PLOS ONE. After careful consideration, we feel that it has merit but does not fully meet PLOS ONE’s publication criteria as it currently stands. Therefore, we invite you to submit a revised version of the manuscript that addresses the points raised during the review process.

**ACADEMIC EDITOR: **Dear authors, the reviewers raised important questions about the manuscript.

You must provide a point-by-point answer to all questions; all changes to the manuscript must be marked in yellow.

We look forward to receiving your revised manuscript.

Kind regards,

Jorddy Neves Cruz

Academic Editor

PLOS ONE

Additional Editor Comments:

Dear authors, the reviewers raised important questions about the manuscript.

You must provide a point-by-point answer to all questions; all changes to the manuscript must be marked in yellow.

Reviewers' comments:

Reviewer's Responses to Questions

**Comments to the Author**

1. Is the manuscript technically sound, and do the data support the conclusions?

Reviewer #1: No

Reviewer #2: Yes

2. Has the statistical analysis been performed appropriately and rigorously? 

Reviewer #1: No

Reviewer #2: Yes

3. Have the authors made all data underlying the findings in their manuscript fully available?

Reviewer #1: No

Reviewer #2: Yes

4. Is the manuscript presented in an intelligible fashion and written in standard English?

Reviewer #1: Yes

Reviewer #2: Yes

5. Review Comments to the Author

Reviewer #1: This manuscript described NMR measurements of serum samples of diabetic model mice (BKS. Cg db/db) at different age groups with increased severity of atherosclerosis. Although control mice (Jc1:ICR) without diabetes was used to bench mark the carotid arteries and lipid profiles, the NMR measurement of control mice was not included. The experiments appear sound, however, the data processing has clear gaps. Thus, the conclusion is not well founded. I would suggest to address the following concerns before consideration for publication.

1. Need to clarify in details of the principal component analysis (PCA), PC1, PC5, Factor-1, Factor-2 and Factor-3. For example, how to reach to PC1 from FID?

2. Suggest to overlay the NMRs of serum samples of mice to identify fingerprint pattern changes over ages as well as over control mice. Or if signal changes subtracting that of control mice has any relationship with ages?

Reviewer #2: The authors used NMR method to analyze serum samples of diabetic model mice and compared the results with the development of atherosclerosis observed in carotid artery samples. They concluded that revealed that NMR modal properties of serum are associated with arteriosclerotic lesions. Thus, it may be worth exploring the possibility that the risk of cardiovascular events in diabetic patients could be assessed using serum samples.

1. Authors should give more examples for the relation of disease or other pathological condition with NMR modes. What are the pathological mechanisms for these relation? Please mention “Nuclear magnetic resonance (NMR) is a phenomenon that occurs when the nuclei of certain atoms are moved from a static magnetic field to an oscillating magnetic field. Radiofrequency pulse at the resonance frequency causes impulse responses so called free induction decays (FIDs) during NMR measurements. By analyzing FIDs using Fourier transform (FT), we can obtain characteristic spectrums of substances (frequency analysis). Some investigators have studied the utility of this modality in discriminating serum taken from patients with cancers such as renal cell carcinoma, hepatocellular carcinoma and lung cancer” from the reference 18. These knowledge is important for readers to understand the topic.

2. “NMR modal analysis of serum is reported useful for the diagnosis of pancreatic cancer”, please explain how it is useful? Am agree the other reviwers that a direct relation seems not logical. On the other hand there may be indirect relation related to inflammatory markers and hormonal fluctuations associated with the onset and progression of hyperglycemia and carotid atherosclerosis

3. Authors said “We recognize that this study has made it possible to identify NMR mode properties associated with carotid atherosclerosis from the serum of diabetic Mouse models.”. Their answer is not based a scientific perspective. Please give more evidence and explanation that are supported with other publications or their hypothesis. Their hypothesis is very weak to understant this relationship.

Minor points

1. Free induction decay (FID should be indicate in the first use.

6. PLOS authors have the option to publish the peer review history of their article (what does this mean?). If published, this will include your full peer review and any attached files.

Reviewer #1: No

Reviewer #2: **Yes: **Bekir Kocazeybek

---

## [Author Response · Author response to Decision Letter 0]

26 Jan 2024

Reviewer #1: This manuscript described NMR measurements of serum samples of diabetic model mice (BKS. Cg db/db) at different age groups with increased severity of atherosclerosis. Although control mice (Jc1:ICR) without diabetes was used to bench mark the carotid arteries and lipid profiles, the NMR measurement of control mice was not included. The experiments appear sound, however, the data processing has clear gaps. Thus, the conclusion is not well founded. I would suggest to address the following concerns before consideration for publication.

1. Need to clarify in details of the principal component analysis (PCA), PC1, PC5, Factor-1, Factor-2 and Factor-3. For example, how to reach to PC1 from FID?

>Response: Thank you for your careful review of our analysis. An explanation of the terms PCA and latent variables (principal components and factors) should facilitate the reader's understanding of the purpose of multivariate analysis in our study. First, we have added an explicit explanation of the significance of PCA in our study to the Methods section.

pp.9, line 195-196. The following sentence has been added: “PCA is a method of dimensionality reduction of data using only explanatory variables and can be used for exploratory analyses such as identifying outliers and potential subgroups [26].”

Next, for the PCA score plots in Figure 3, we were only concerned with explaining PC-1. From our exploratory study, it is possible to explain the group information, although the biological or chemical interpretation of the principal components is difficult. Therefore, we have added an explanation of the group information combining PC-1 and PC-5 to the Results section.

pp.14, line 293-296. The following has been added: “Principal component 1 score was interpreted to be positively associated with week-old progression, whereas principal component 5 score was interpreted to be negatively and moderately associated with week-old progression. Furthermore, on closer examination of principal component 1, it shows …..”

Furthermore, in this study, PLS-DA was used as a two-class classifier. Thus, Factor-1 of the PLS-DA in Figures 4 through 7 is interpreted as an axis that reflects group information; Factor-2 and Factor-3 are significant for maximizing separation on Factor-1 and providing axes that make this visible, but have no specific biological interpretation.

pp.15, line 314-318. The following sentence has been added: (Figure 4B displays …. NMR signal.) “Because PLS-DA was used as a 2-class classifier in the present study, Factor-1 in Figure 4 is interpreted as an axis that reflects group information; axes from Factor-2 onward are interpreted as axes that maximize the separation on Factor-1 and make it visible on the plot. This is also the case for Figures 5 through 7. This interpretation applies to Figures 5 through 7 as well.”

Finally, we consider that the meaningful question "how to reach PC1 from FID?" suggests that readers may have difficulty understanding the analysis process in this paper. For readers unfamiliar with this area, a concise diagram would facilitate understanding of the method. Therefore, we have added a graphical flow chart that focuses primarily on the analysis process in this paper. In this supplemental chart, we have added the articles on PCA, PLS-DA, and STFT to the legend so that readers who wish to review specific methods can do so quickly.

pp.9, line 209-210. Supporting figures and their legends have been added, and the following sentence has been inserted: A graphical flow chart from animal experiments to the multivariate analysis performed in this study was shown as supporting information.

pp.27, line 603-625.

“Supporting information

S1 Fig. Graphical flow chart from the animal experiments to the multivariate analysis

In this figure, the results of NMR analysis of blood samples collected after animal experiments in groups A (n=10) and B (n=10) are shown. In total, 20 1H-NMR FID signal data were acquired, which were short-time Fourier transformed [51], and 20 spectrograms were calculated. For multivariate analysis, a process was performed to obtain a data matrix from the 10 spectrograms. From this data matrix, an exploratory analysis was performed using principal component analysis [26] for the identification of outliers and potential subgroups of animals in groups A and B. From this figure, it was possible to identify subgroups that were not initially anticipated, so PLS-DA [52,53] was performed only with data that corresponded to subgroups in the same data matrix to analyze whether subgroups could be identified. In both principal component analysis and PLS-DA, the variables that were important for the identification of groups in each analysis were plotted as corresponding score plots. This score plot has the same time and frequency axes as the spectrogram and thus can be compared with the spectrogram. The illustration of the mouse and the NMR equipment in this figure was obtained from Research Net, a website where research illustration material is available free of charge (https://www.wdb.com/kenq/illust © WDB Co., Ltd., Tokyo, Japan).

S2 Fig. Graphical flow chart of matrixing of spectrogram data

The process surrounded by the green box in Supporting figure 1 is described in detail. For multivariate analysis of spectrogram data, a 256 × 1024 spectrogram (a) with frequency on the horizontal axis and time on the vertical axis was divided into 1024 single rows (b). All rows were re-arranged into a single row, and 1 × 262,144 single rows were reshaped (c). Since a single row represents spectrogram data for one individual (d), the single rows of spectrogram data for all individuals needed for multivariate analysis are combined to create a data matrix (e).”

2. Suggest to overlay the NMRs of serum samples of mice to identify fingerprint pattern changes over ages as well as over control mice. Or if signal changes subtracting that of control mice has any relationship with ages?

>Response: Thank you for your valuable comments. We would like to draw your attention to the score plot that was presented only to the peer reviewers (re-presented below). In this plot, the control and diabetic groups are clearly separated: the variation in the 18-week-old control group is greater than the variation in the 10- to 26-week-old diabetic group, and there are two individuals who are outliers. While this may include an element of strain differences, the results can be interpreted as indicating that the plots of experimental groups with pre- or post-diabetes and atherosclerosis tend to converge more than plots of normal, but with inevitable variation, "control" animals.

[Please see the figure in the accompanying Response to Reviewers file.]]

The identification of signal components in control animals to delineate the boundary between normal and abnormal is one of the most important procedures in bioinformatics research. However, the large variability in the NMR modal properties of the control mice presented in this study makes the process of superposition and subtraction of NMR signals extremely difficult to interpret. Furthermore, age has been noted as one of the intrinsic physiological factors affecting the composition of biological samples from laboratory animals [Bollard ME, Elizabeth G Stanley, John C Lindon, Jeremy K Nicholson, Elaine Holmes. NMR-based metabonomic approaches for evaluating physiological influences on biofluid composition. NMR Biomed. 2005 May;18(3):143-62. PMID: 15627238 DOI: 10.1002/nbm.935]. Thus, the loading plot in Figure 3B can be considered to show NMR modal properties associated with diabetes and atherosclerosis that develop with age.

In conventional spectral analysis, spectral features associated with age or control animal groups may be considered as NMR fingerprint patterns. However, the short-time Fourier transform applied in this study requires arbitrary optimization of the analysis conditions, especially the window function settings, to meet the analysis objectives. Therefore, showing a specific fingerprint pattern from the NMR spectrogram may lead to a great deal of misinterpretation by the reader.

Though the reviewers have provided valuable comments, unfortunately, due to the nature of the STFTs used in this study, we would appreciate your understanding that we are not including the fingerprint pattern changes related to other ages, as well as other control mice. 

Reviewer #2: The authors used NMR method to analyze serum samples of diabetic model mice and compared the results with the development of atherosclerosis observed in carotid artery samples. They concluded that revealed that NMR modal properties of serum are associated with arteriosclerotic lesions. Thus, it may be worth exploring the possibility that the risk of cardiovascular events in diabetic patients could be assessed using serum samples.

1. Authors should give more examples for the relation of disease or other pathological condition with NMR modes. What are the pathological mechanisms for these relation? Please mention “Nuclear magnetic resonance (NMR) is a phenomenon that occurs when the nuclei of certain atoms are moved from a static magnetic field to an oscillating magnetic field. Radiofrequency pulse at the resonance frequency causes impulse responses so called free induction decays (FIDs) during NMR measurements. By analyzing FIDs using Fourier transform (FT), we can obtain characteristic spectrums of substances (frequency analysis). Some investigators have studied the utility of this modality in discriminating serum taken from patients with cancers such as renal cell carcinoma, hepatocellular carcinoma and lung cancer” from the reference 18. These knowledge is important for readers to understand the topic.

>Response: Thank you for presenting a topic that contains thought-provoking perspectives in this paper. We would like to express our appreciation. As pointed out, Sato et al. mention that “Some investigators have studied the utility of this modality in discriminating serum taken from patients with cancers such as renal cell carcinoma, hepatocellular carcinoma and lung cancer.”[19] These studies, like our experiments, used proton nuclear magnetic resonance (1H NMR) spectroscopy and human serum samples. On the other hand, unlike our NMR modal analysis method, their studies were NMR metabolomics. In these papers, specific metabolites were identified, offering the prospect that they may be useful for early diagnosis of malignant disease, identification of advanced subgroups, and postoperative surveillance.

Although research on the association between NMR modal analysis methods and diseases or pathological conditions is ongoing, only research on pancreatic cancer has been published as a scientific article at this time [19]. Therefore, unfortunately, we cannot provide specific, biomedical examples of these associations.

Therefore, after careful consideration of the reviewer’s comments, we decided that it was appropriate to explain the pathological mechanisms for these relations from the principles of NMR. The NMR measurements and their analysis in this study are performed on mixtures of low molecular weight materials such as metabolites in biological samples and high molecular weight materials such as serum proteins, without deproteinization in sample preparation or NMR measurements such as T2 filtering. In the method for mixture analysis, Dumez [Ref. 45] points out that “in spectra of mixtures, such as liquid samples or extracts of biological samples, signal overlap prevents reliable identification and accurate quantitation of signals.” His review focuses on homogeneous solutions and on small molecules, which is also a concept in NMR metabolomics. On the other hand, we focus non-selectively on all molecular entities in a biological sample, including protons, which makes the NMR modal analysis method innovative in its ability to comprehensively analyze the information that a biological sample contains.

[Please see the table in the accompanying Response to Reviewers file.]

It is assumed that the NMR mode of biological samples from a group of diseases is affected by biochemical substances (hormones, inflammatory substances, etc.) specific to that disease. As for the pathological mechanism, we believe that, in addition to the abnormal metabolites produced by the substances, the physical properties of macromolecular substances such as albumin, which are affected by the substance profiling produced in this way, also have a significant impact. This attention to this point is a major difference between NMR modal analysis and NMR metabolomics, and in light of the differences with NMR metabolomics, we have added the following text to facilitate understanding of the pathological relevance of NMR modal properties.

pp.19, line 419-427. The following text has been added: “NMR metabolomics experiments focus on low molecular weight substances and obtain their spectra [45]. On the other hand, in NMR modal analysis, all molecular species including protons of up to high molecular weight substances such as serum proteins in addition to low molecular weight metabolites are measured by NMR in a non-selective and comprehensive manner to obtain spectrograms that reflect the state of all hydrogen atoms in the serum sample. This allows us to interpret that biochemical changes due to disease or pathological abnormalities may be reflected in not only static but also dynamic changes of protons in all components in serum (from low to high molecular weight substances and free water). In this study,…..”

2. “NMR modal analysis of serum is reported useful for the diagnosis of pancreatic cancer”, please explain how it is useful? Am agree the other reviwers that a direct relation seems not logical. On the other hand there may be indirect relation related to inflammatory markers and hormonal fluctuations associated with the onset and progression of hyperglycemia and carotid atherosclerosis

>Response: Our intention was to present the NMR modal method as a previous study applied to clinical specimens, but our wording was misleading. As noted, we also considered that “there may be an indirect relation related to inflammatory markers and hormonal fluctuations associated with the onset and progression of hyperglycemia and carotid atherosclerosis”. Therefore, to ensure that our intentions are accurately conveyed, we have revised the text as follows.

pp.4, line 72-76. The following sentence has been added: “As an example, there are published studies in which NMR modal analysis of serum from pancreatic cancer patients yielded results of diagnostic importance [19]. Therefore, we considered the possibility of applying our method not only to pancreatic cancer, but also to other diseases that indirectly or directly alter the physical properties of serum, specifically, the mobility of all hydrogen nuclei contained in serum. In the present study,….”

3. Authors said “We recognize that this study has made it possible to identify NMR mode properties associated with carotid atherosclerosis from the serum of diabetic Mouse models.”. Their answer is not based a scientific perspective. Please give more evidence and explanation that are supported with other publications or their hypothesis. Their hypothesis is very weak to understand this relationship.

>Response: As pointed out, we recognize that our statement that this study has made it possible to identify NMR mode associated with carotid atherosclerosis from the serum of diabetic mouse models is somewhat overstated. Considering the results presented by our experiments, our response to this comment is revised based on the results of our experiments as described below: “It is suggested that this study has made it possible to identify NMR mode associated with carotid atherosclerosis from the serum of diabetic mouse models.”

 Furthermore, in response, we have also added or revised the text as follows. As pointed out by the reviewers, we have responded from a scientific perspective by adding evidence and explanations supported by other papers and their hypotheses.

Data Analysis Strategy

The purpose of our study was to provide proof-of-concept that the development of atherosclerosis in individuals with a diabetic background can be diagnosed from serum NMR modal experiments. In other words, we consider that identifying serum NMR modes associated with atherosclerotic lesions was key in this experiment. This is essentially different from metabolomics experiments [Alonso-Moreno P, Rodriguez I, Izquierdo-Garcia JL. Benchtop NMR-Based Metabolomics: First Steps for Biomedical Application. Metabolites 2023, 13(5), 614; https://doi.org/10.3390/metabo13050614], which aim to identify biomarkers and understand the underlying metabolic changes associated with various diseases. As in the pancreatic cancer paper, the NMR modal method step in this paper used pattern recognition methods [25]. Since the purpose of this experiment was identification, a combination of PCA and PLS-DA was used.

pp.8-9, line 188-191. The following sentence has been added: “Similar to the study of the diagnosis of pancreatic cancer from serum by applying NMR modal methods [19], pattern recognition [25] of image data was applied to clarify groups from 1H NMR-FID spectrograms of mouse serum in this study, and principal component analysis (PCA) and partial least squares discriminant analysis (PLS-DA) were performed.”

pp.9, line 191-192. The underlined words have been revised as followed: “To perform PCA and PLS-DA, …”

Animal experiments

In order to achieve the goal of identifying atherosclerotic lesions from serum in our experiments, it is crucial that the experiments are carefully planned and well controlled. This is also true for metabolomics experiments, as Ebbels et al. point out [Ref 31], “One of the most important stages, but one which is often overlooked, is the initial design of the experiment.” As reviewer #1 noted, "The experiments appear sound," and we are confident that the animal experiments have been completed successfully enough.

 In addition, we performed typical blood biochemistry experiments such as blood glucose, T-cho, TG, etc., as shown in Table 2, but although blood biochemistry results varied from individual to individual, this variation in results did not clearly correlate with the scatter in the PCA score plots in Figure 3.

pp.17, line 368-373. The following sentence has been added: “It can be pointed out that one of the factors contributing to this success was the successful completion of animal experiments conducted under well-controlled conditions, and with the exception of one individual, pathology experiments on mouse carotid arteries were also successful. In fact, it has been pointed out that the first animal experiment is a very important element in the interpretation of the results of multivariate analysis in the step of chemometric studies [31].”

pp.14, line 298-301. The following sentence has been added: “Furthermore, although the blood biochemistry test results shown in Table 2 varied from individual to individual, there is no clear correlation between this variation and the variation in the PCA score plots in Figure 3.”

Principal Component Analysis

If the PCA in Figure 3 formed subgroups by age, the majority of the NMR modes of mouse serum could be interpreted as being influenced by age. However, the 18-week-olds were separated into two groups, and the NMR modes of mouse serum cannot be simply classified by age alone.

 To assist in the interpretation of the factors contributing to the separation of the two groups of 18-week-old mice in Figure 3, carotid artery pathology was used in this experiment. Carotid artery pathology was applied to specimens from all ages, not just the 18-week-old individuals, so that lesion evolution could also be confirmed. The results, as shown in Table 1, indicate that 18-week-old mice are in a period of rapid development of carotid atherosclerosis. This result makes us suggest that carotid atherosclerosis may have influenced the distribution of the score plots, although the PCA in Figure 3 may have been affected by aging to a greater or lesser extent.

pp.17, line 374-381. The underlined word and sentences have been added as followed: “Based on the score plots of all PCA data for BKS.Cg db/db mice, all groups (except B18) were divided into a precursor group and progression group by PC-1. In contrast, the data plots for group B18 were scattered and bifurcated into precursor and progression groups, as the atherosclerotic lesions progressed particularly rapidly. If the scatter of the score plots in Figure 3 reflected differences in week age, a distribution by age group would be expected, but the actual results are contrary to this. Since PCA is an analysis that uses only explanatory variables (in this experiment, the serum spectrogram data) for dimensionality reduction [26], it was thought that the results could be interpreted as a visualization of changes in serum status around 18 weeks of age.”

PLS-DA and loading plot

Next, based on the results of carotid artery pathology in 18-week-old mice, the PCA results were divided into several subgroups, and the PLS-DA was used for their classification. The clarity of the separation is shown in Figures 4-7, but it is difficult to determine whether the separation is meaningful or not only from the score plots.

 Therefore, a new tool we have prepared for this study is a two-dimensional loading plot based on the spectrogram. This allows us to visualize which areas on the spectrogram are effective pixels for identification. Brereton et al. pointed out the following [Richard G. Brereton, Gavin R. Lloyd.　Partial least squares discriminant analysis: taking the magic away. J.Chemometrics 2014;28(4):213-225. https://doi.org/10.1002/cem.2609]; As an exploratory graphical method that suggests which variables are the most likely to be responsible for discrimination, used often in exploratory studies, PLS-DA has a significant role to play. If the result is that PLS-DA is effective in discriminating in areas such as noise regions where the energy intensity is very weak on the spectrogram, then the results in Figures 4-7 must be interpreted as not particularly meaningful. However, as shown in Figures 4B-7B, the loading plots confirm that the regions correspond to the strong energy regions of the spectrogram as effective regions for identification. This can be interpreted as suggesting that the PLS-DA discrimination in this experiment may be due to differences in serum NMR mode. This step is equivalent to performing a PLS-DA in a metabolomics study to produce a score plot and a corresponding (one-dimensional) loading plot.

pp.17, line 384. The underlined words have been inserted as followed: “…part of these samples was classified into the precursor group. As shown in Fig. 7A, the results for groups B10, B14, and B18 A showed that the samples were divided into these groups by Factor-1.”

pp.17-18, line 387-393. The underlined sentences have been added as followed: “The PLS-DA score plot for this precursor group shown in Figure 7A cannot be explained by age-related changes alone. In this study, time-trend changes in carotid artery pathology were observed, which, as noted above, showed rapid atherosclerotic lesion progression at 18 weeks. Considering the characteristics of the distribution of the B18 group shown on some score plots, we interpret this as suggesting that the clustering and separation of samples shown in the multivariate score plots in this study may be related to the progression of atherosclerotic lesions, though there is a certain degree of influence of differences in age.”

pp.18, line 394-397. The underlined sentences have been added and revised as followed: “In addition, the time-frequency domains of the variables valid for discrimination shown in the loading plots in Figures 3-7 corresponded roughly to the signal domains in the serum spectrograms in Figure 2. These results suggest that the progression of arteriosclerotic lesions was reflected in the serum NMR modal properties.”

Interpretation of NMR mode

The purpose of this experiment was different from that of metabolomic analysis, so we will not identify the metabolite responsible for the identification. On the other hand, we conducted carotid artery pathology examinations in mice and interpreted the NMR mode based on the histopathology findings.

 The changes in carotid artery pathology at and around 18 weeks of age naturally lead us to infer a pathophysiological mechanism by which the properties of mouse serum would be affected by inflammatory changes and hormonal fluctuations during this period. We interpreted this change in blood properties, either directly or indirectly, as a major variation in serum NMR mode at 18 weeks of age. Because the present experimental data, unlike metabolomic analysis, are of interest for all molecular species (low to high molecular weight substances and free water), including protons in serum, and can be regarded as reflecting molecular species and their amounts, as well as intra- and intermolecular forces acting on the protons in each molecule, the NMR modal analysis method has the potential to sensitively observe differences in the physical properties of the serum as a whole. Based on appropriately controlled animal studies, we hypothesize that the differences in NMR modal properties shown in this study are due to mouse carotid atherosclerosis.

Since the content is related to reviewer #2's comment 1, the addition of the manuscript is the same as in “pp.19, line 419-427” above. 

pp.19, line 419-427. The following sentence has been added: “NMR metabolomics experiments focus on low molecular weight substances and obtain their spectra. On the other hand, in NMR modal analysis, all molecular species including protons of up to high molecular weight substances such as serum proteins in addition to low molecular weight metabolites are measured by NMR in a non-selective and comprehensive manner to obtain spectrograms that reflect the state of all hydrogen atoms in the serum sample. This allows us to interpret that biochemical changes due to disease or pathological abnormalities may be reflected in not only static but also dynamic changes of all protons in components in serum (from low to high molecular weight substances and free water).”

Issues in the future

According to Bollard et al, "Each of the spectral integrals can be regarded as a metabolic descriptor which can be used to investigate similarities or differences in the data, according to an individual's biochemistry.” Namely, in metabolomics research, spectrum-based loading plots and NMR spectral fingerprints can be matched to interpret metabolic changes associated with various diseases.

 On the other hand, the physicochemical meaning of the calculations of the NMR spectrograms will be clarified in further studies. The next question is what kind of biomedical interpretation of spectrogram-based loading plots will be possible once the meaning of the spectrograms is clarified. Furthermore, there are several issues to be addressed, including room for improvement in statistical methods, larger sample size for model building, the relevance of NMR mode properties to diseases and pathologies, and the applicability of NMR to more complex human clinical specimens. However, despite some of the challenges, the current paper is positioned as a feasibility study for future serum NMR modal analysis and diagnosis of carotid artery extension in more complex human blood specimens.

The above response is meant as an explanation to the reviewers and does not result in revisions or additions to the manuscript. 

Minor points

1. Free induction decay (FID should be indicate in the first use.

pp.7, line 142. The underlined words have been revision where FID was used for the first time: “Free induction decay (FID) data collection for NMR measurement…”

References

For changes in reference numbers made to accommodate the addition of references, please see the table in the accompanying Response to Reviewers file.

---

## [Decision Letter · Decision Letter 1]

13 Feb 2024

Time-frequency analysis reveals an association between the specific nuclear magnetic resonance (NMR) signal properties of serum samples and arteriosclerotic lesion progression in a diabetes mouse model

PONE-D-23-28528R1

Dear Dr. Kanawaku,

We’re pleased to inform you that your manuscript has been judged scientifically suitable for publication and will be formally accepted for publication once it meets all outstanding technical requirements.

Kind regards,

Jorddy Neves Cruz

Academic Editor

PLOS ONE

Additional Editor Comments (optional):

Reviewers' comments:

Reviewer's Responses to Questions

**Comments to the Author**

1. If the authors have adequately addressed your comments raised in a previous round of review and you feel that this manuscript is now acceptable for publication, you may indicate that here to bypass the “Comments to the Author” section, enter your conflict of interest statement in the “Confidential to Editor” section, and submit your "Accept" recommendation.

Reviewer #1: All comments have been addressed

Reviewer #2: All comments have been addressed

2. Is the manuscript technically sound, and do the data support the conclusions?

Reviewer #1: Yes

Reviewer #2: Yes

3. Has the statistical analysis been performed appropriately and rigorously? 

Reviewer #1: N/A

Reviewer #2: Yes

4. Have the authors made all data underlying the findings in their manuscript fully available?

Reviewer #1: Yes

Reviewer #2: Yes

5. Is the manuscript presented in an intelligible fashion and written in standard English?

Reviewer #1: Yes

Reviewer #2: Yes

6. Review Comments to the Author

Reviewer #1: The patern learning is an interesting way to explain the data but the direct molecular bases need to be established before it can be useful. Encourage to do further studies and communicate later as a followup.

Reviewer #2: All revisions were performed and the manuscript can be understandabl now, in this current form.T he authors also respond my comments in a scientific perspective.

7. PLOS authors have the option to publish the peer review history of their article (what does this mean?). If published, this will include your full peer review and any attached files.

Reviewer #1: No

Reviewer #2: **Yes: **Bekr Kocazeybek

---

## [Editor Report · Acceptance letter]

26 Feb 2024

PONE-D-23-28528R1 

PLOS ONE

Dear Dr. Kanawaku, 

I'm pleased to inform you that your manuscript has been deemed suitable for publication in PLOS ONE. Congratulations! Your manuscript is now being handed over to our production team.

Kind regards, 

on behalf of

Dr. Jorddy Neves Cruz 

Academic Editor

PLOS ONE